# Measuring and Mapping Physical Activity Disparity (PAD) Index Based on Physical Activity Environment for Children

Jue Yang [1], Lan Mu [1,*] and Janani Rajbhandari-Thapa [2]

1   Department of Geography, University of Georgia, Athens, GA 30602, USA; tvjoyxq@uga.edu
2   Department of Health Policy & Management, University of Georgia, Athens, GA 30606, USA;
    jrthapa@uga.edu
*   Correspondence: mulan@uga.edu

**Abstract:** Physical activity (PA) plays a vital role in children's physical and mental health. The built, natural, and socio-demographic environmental variables affect children's PA behaviors in various ways. However, few studies focus on systematically measuring the environmental spatiality to enhance PA research. We propose a Physical activity Access Disparity (PAD) index for children. This study aims to design, test, and apply an integrated approach to the children's PAD index. We adopt five dimensions of "access" to healthcare to measure the children's PAD index for the United States (US) and the state of Georgia at the county level. The PAD index sorts 18 environmental measures with 23 variables into accessibility, availability, accommodation, affordability, and acceptability (5 As) for children's PA. We use the self-organizing map (SOM) method to measure how the 5 As affect the PAD index values. According to the result, the children's PAD index's ranking normalizes from 0 to 1 and identifies "play oases" to "play deserts" in the US and Georgia using diverse 5 As combinations. The children's PAD index shows Low disparity in the north and coastal region and High disparity in Deep South states in the US. Moreover, the PAD index shows Low disparity and High disparity in the north and south of Georgia. The PAD index provides a valuable tool for researchers and policymakers to analyze disparity in children's "access" to the PA environment. The flexible parameters and the weighing scheme also extend the method's generality and allow users to customize the PAD index based on local preferences and conditions.

**Keywords:** children; physical activity environment; Physical Activity Disparity (PAD) index; GIS; self-organizing map

## 1. Introduction

The lack of physical activity (PA) has become one of the leading risk factors for chronic health conditions, and preventable deaths in the United States (US) and the world [1,2]. Particularly among children (under 18) [3], PA is associated with a positive effect on overall health. Benefits include improved attention and memory, strengthened bones, strong muscles, endurance, reduced obesity, and more [4]. According to WHO [1], three-fourths of children globally have PA rates lower than recommended. These statistics demonstrate the need to study the determinants of low rates of PA. Numerous studies have suggested that investigating the association between the environment and PA could guide intervention strategies to improve PA among children [5–7].

Environmental factors affect children's engagement in PA [8,9]. Such factors can be classified as built, socio-demographic, and natural environmental factors. For instance, Brownson [8] categorized features in the built environment that influence PA into five domains: functional, safety, aesthetic, destination, and other. These environmental variables have been widely discussed individually in literature related to children's access to the PA environment, such as the availability of park features, accessible recreation facilities, free open space, and neighborhood safety [7–11]. However, few papers have examined

the relationship between multiple measurements of PA environment and children's access to PA opportunities. For example, a park includes features such as park accessibility, availability, free vs. paid access, park safety, and park aesthetics and more to explain how these features affect the PA environment of the park and access to PA opportunities among children. Several park attributes establish the accessibility and usability of the park, and, in turn, attributes of PA determine children's access to PA opportunities [11,12]. Furthermore, the PA environment differs across various geographical levels, such as county, state, and region. There is also a need for analysis to expand to various locations. This study tries to fill the gap in the current literature by creating a comprehensive multi-dimensional access framework for children's PA environmental variables among different geographical levels. We adopt five measures of access, namely availability, accessibility, affordability, accommodation, and acceptability (commonly referred to as the 5 As of access) and Geographic Information System (GIS) methods to design and build a children's Physical Activity Disparity (PAD) index for the US. We then demonstrate the feasibility of the children's PAD index framework to adapt to a finer geographical level, such as states in the case of US, based on data availability and state policy priorities.

Our research questions are: What is the range of "play oases" to "play deserts" in the US and Georgia followed by the 5 As framework? How do the 5 As, individually or in combination, impact the physical activity disparity? The present study is structured as follows: Section 2 provides the background of the PA environment for children and the 5 As' concept to develop the children's PAD index. Section 3 describes our data analysis methodology, which encompasses data collection, the design of the PAD index for children, and the application of the Self-Organization Map to classify the PAD index. Section 4 presents a summary of the findings from our study. Section 5 offers a discussion of the contributions of the PAD index and suggestions for future research. Finally, in Section 6, we provide a conclusion to the article. The List of acronyms is in Table S1.

## 2. Background

### 2.1. Children's PA Environment

The PA environment is a space where PA occurs, such as green spaces, health clubs, sidewalks, and trails [13]. Many works of literature commonly focus on PA behaviors. For instance, Pontin et al. [14] used a smartphone app to gain new insights into the socio-demographic variation in PA behaviors; Mitchell [15] indicated that the natural environmental variable affects human PA to produce mental health benefits. Such behavior is not only limited to behavior but also depends on the PA environment [13,16]. The environmental variables are even more important in the context of children's PA as they have restricted access and must be accompanied by an adult. Therefore, the PA environment that children have access to is an essential predictor of children's PA behavior [17]. For example, Flaes et al. [18] conducted a study that indicated that access to a better quality park environment increased PA (i.e., energy expenditure) among children. Improved access to PA environment (e.g., walkability, playability, accessibility, and safety) among children is a vital strategy to influence children's health outcomes [19].

### 2.2. The 5 As of Access to Children PA Environment

Penchansky and Thomas [20] proposed the 5 As of the access concept to study the degree of fit between characteristics of healthcare supply and population demand. The supply and demand characteristics include objective characteristics such as count, socio-demographics, and transportation, and subjective characteristics such as people's attitudes or relationships. Penchansky and Thomas grouped those characteristics into five dimensions of access: Availability, Accessibility, Accommodation, Affordability, and Acceptability [20]. Availability measures the supply of services in relation to demand. Accessibility highlights the geographical linkage between supply and demand, considering travel distance, time, and cost. Accommodation refers to the supply of resources organized to meet the demand preferences, including the waiting time or application procedures. Affordabil-

ity measures the demand, the population's ability to afford the available supply of resources or insurance coverage. Acceptability also reflects demand in terms of the population's attitude towards the supply, such as gender, culture, and ethnicity [21,22].

We adopt the 5 As' concept in this paper for the PA supply to represent what the environment and society can offer in terms of children's PA, and its counterpart is the children's demand for PA. The availability and accessibility of PA facilities support the opportunities for children's PA [23]. Low poverty levels and income tend to impact people's access to free open space [24]. Steep hills decrease the frequency of outdoor activity [10], and so do safety concerns for outdoor activity and active transportation [25]. Therefore, there is a need to illustrate the 5 As' framework for children's PA [26]. With the 5 As' multi-dimensional framework, environmental variables could be remeasured to fit the framework which added a new insight into PA environmental research.

## 3. Data and Method

### 3.1. Designing a Children's PAD Index

For the development of the children's PAD index, we collected several potential PA measures from built, natural, and socio-demographic environments and classified characteristics of measures, which are referred to as variables, into the 5 A dimensions. The measurement was based on the variable with Multi-Criteria Evaluation (MCE) to calculate the PAD index [27]. We design and present the children's nationwide PAD at the county level for US. To show the flexibility and customization in building the PAD index, we collected environmental measures statewide in Georgia (GA) and demonstrated the flexibility of the 5 As in designing the PAD index at a regional level. Lastly, we analyzed how 5 As influence the level of the PAD index by using the self-organizing map (SOM) method.

### 3.2. Environment Measures Selection

To identify the potential environmental measures within the PAD index, we reviewed the literature using keyword combinations of "children", "adolescents", "physical activity", "recreation", "built environment", "socio-demographic" and "natural environment". Ten review articles focused on one or more elements of children's PA environmental measures [7,8,28–35]. According to the literature and data availability, we chose 15 measures (No. 1–7, 10–13, and 15–18 in Table 1) for the national analysis and 18 measures (No. 1–18 in Table 1) for the state (GA) analysis, as listed in Table 1. Some measures have more than one variable. For example, the park variable (No. 7 in Table 1) includes park count, nearest distance to the park, open access park proportion, and park activity diversity (see Section 3.3 for the explanation of the measures under each of the 5 As' domains). The study proceeded with 18 variables (in 15 measures) for the national analysis and 23 variables (in 18 measures) for the state analysis (Figure 1). The five additional variables indicate the variables that were available only at the state (GA) level.

**Table 1.** Environmental measures description.

| No. | Measures | Data Source | Description |
|---|---|---|---|
| | | Built Environment | |
| 1 | Business Recreation | Safegraph [36] | The total count of the following NAICS business locations refers to the business recreation points related to the children's activity. The business locations include Recreational and Vacation Camps; Fitness and Recreational Sports Centers; All Other Amusement and Recreation Industries; Zoos and Botanical Gardens; Amusement and Theme Parks; Sports and Recreation Instruction (under education); Museums; Amusement Arcades; and Historical Sites. |

**Table 1.** *Cont.*

| No. | Measures | Data Source | Description |
|---|---|---|---|
| 2 | School | National Center for Education Statistics [37] | Public Elementary, Middle, and High Schools locations |
| 3 | National walkability Index | US Environmental Protection Agency [38] | National Walkability Index score To determine the walkability scores, the intersection density, land use mix, and proximity to transit were integrated into the score calculation. |
| 4 | Road density | ArcGIS Online [39] | Roads, highways, bike trails, and footpaths. |
| 5 | Crime Rate | United States Department of Justice [40] | Uniform Crime Reporting Program Data |
| 6 | Vacant place | Census American Community Survey (ACS) [41] | Vacant housing units |
| 7 | Park | ArcGIS Online | Public parks, gardens, and forest polygons |
|   |   | Georgia Department of Parks & Recreation (GDPR) [42] | Park points with a set of 13 core activities desired and park status by GDPR |
| 8 | Business Review Rate | Google Maps reviews [43] | Average Google map review Rate for Business Recreation point |
| 9 | Traffic | Georgia Department of Transportation (GDOT) [44] | GDOT Traffic count point |
| Natural Environment | | | |
| 10 | Slope | US Geological Survey [45] | North America elevation 1 km resolution raster |
| 11 | Greenness (NDVI) | MOD13A2 Version 6 [46] | The average NDVI raster from 2021 May to 2021 October |
| 12 | Air pollution (PM 2.5) | US Environmental Protection Agency [47] | PM 2.5: the values shown are the highest among the sites in each county and the weighted annual mean concentration. |
| 13 | National Risk Index | Federal Emergency Management Agency [48] | The index illustrates the US communities most at risk for 18 natural hazards at the county level. It includes avalanches, coastal flooding, cold wave, drought, earthquake, hail, heatwaves, hurricanes, ice storms, landslides, lightning, riverine flooding, strong wind, tornado, tsunami, volcanic eruption activity, wildfires, and winter weather. |
| 14 | Water pollution | Environmental Protection Division of Georgia (GA) [49] | Water pollution point (Water pollution categories: 4a, 4b, 4c, 5 and 5R) |
| Socio-demographic Environment | | | |
| 15 | Income | Census ACS [41] | Household income |
| 16 | Below Poverty | Census ACS [41] | Population below the poverty level |
| 17 | Unemployment Rate | Census ACS [41] | Unemployment rate |
| 18 | No High School Diploma | Census ACS [41] | Completion or high school which is equivalent to the not completing post-secondary non tertiary education according to the international standard of education |

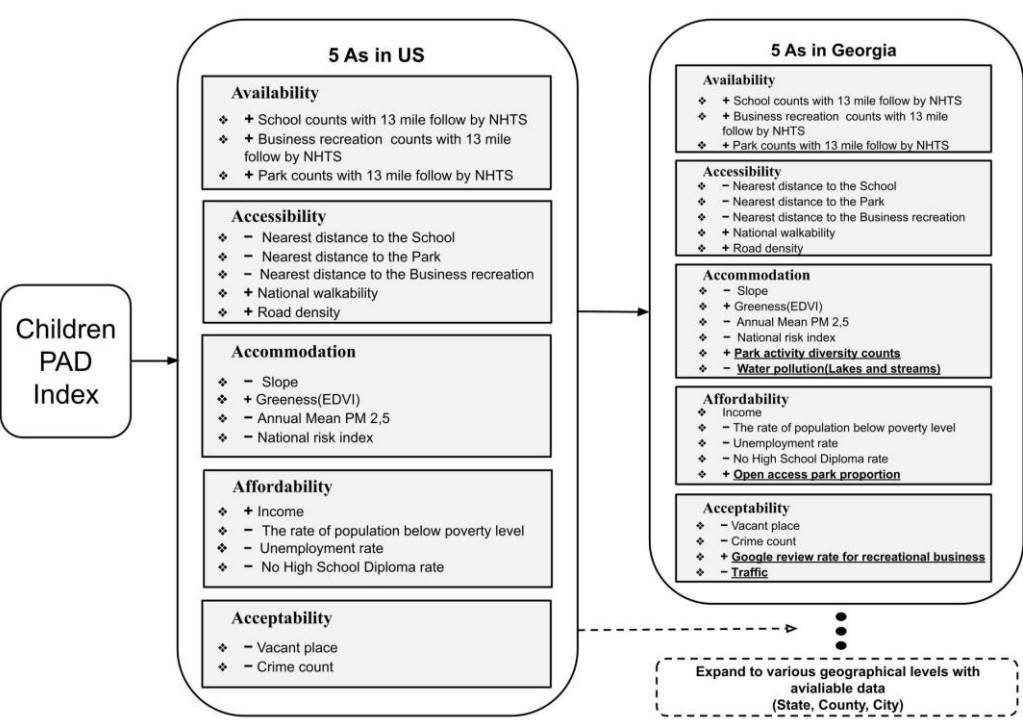

**Figure 1.** The variable distribution of 5 As for Children's PAD indexes in the US and Georgia.

### 3.3. Environmental Measures' Variables for 5 As in the US and Georgia

For all the variables we included in the 5 As calculation, we aggregated and averaged them to the county level.

**Variables of Availability**: Availability defines the adequacy of the PA supply. The counts of business recreations, schools, and parks were defined as the PA supply with a positive direction. A larger supply count value for these three variables means higher availability (Figure 1). We used 13 miles (20.9 km), the average travel distance to recreation destinations for both the US and Georgia, according to the National Household Travel Survey's [50] recreation travel distance mode sample, to define the area for the PA supply count. We created 13-mile Euclidean distance buffers, which calculated the direct distance from the central points of each block group polygon and calculated the PA supply count within each buffer. Then, the PA supply counts of block groups were aggregated to obtain the mean value at the county level. Finally, the population-adjusted PA supply per 1000 children was calculated to generate the Availability index in the US and Georgia.

Variables of Accessibility: Accessibility demonstrates the population's trip toward the available PA supply. The accessibility variables included the nearest distance to PA supply, walkability, and road density. The nearest distance to a business recreation, school, and park indicated that a shorter distance between the population location, as the block group centroid, and PA supply's location showed higher accessibility. Higher walkability and higher road density show higher accessibility. Therefore, these five variables were calculated to generate the Accessibility index in the US and Georgia.

**Variables of Accommodation**: Accommodation indicates how the PA supply accommodates the children in the objective circumstances. Our study defined the index from the environmental aspect to see whether the PA environmental conditions, such as nature characteristics (river and mountain), facilities, and amenities accommodate children's demands. The four measures, slope, greenness (normalized difference vegetation index; NDVI), air pollution, and National Risk Index (NRI) were collected and reclassified at the county level. Higher values of the slope, NRI, and annual mean PM2.5 indicated a lower accommodation, and higher greenness values indicated a higher accommodation. These four variables were used to generate the Accommodation index in the US. In Georgia, we

added two more variables, i.e., water pollution and park facility diversity, to analyze the Accommodation index of Georgia.

**Variables of Affordability**: Affordability demonstrates the affordability of the PA supply. Here, we included the variables that define the socioeconomic status of children that directly affects the demand for PA. It includes the household income, poverty level (being below the 10.5 percent national poverty level in the 2019 US Census standard), unemployment rate, and no high school diploma rate. Higher income indicated higher affordability. Below-poverty-level rate, unemployment rate, and no high school diploma rate indicated low affordability. These four variables were used to generate the Affordability index in the US. In Georgia, we added the public park rate for all parks as high affordability for the expense characteristics of the supply to analyze the Affordability index of Georgia.

**Variables of Acceptability**: Acceptability shows the attitude towards the safety and usability of the PA supply. Our study explained it as the children's or guardians' accepted environment for the children to play within comfortably and safely. Crime and vacant lot counts within each county are used to measure the Acceptability index in the US. A higher count indicates lower acceptability. In Georgia, the traffic data and average rate of Google Maps reviews for the business recreation supplies were added to the Acceptability index of Georgia.

*3.4. As Index Measurement*

To design the children's PAD index, we classified the variables of the environmental variables into 5 As (Figure 1). The Increase (+) or Decrease (−) signs illustrate the variables' contribution to the PAD index. To avoid a calculation error, we followed the contribution direction in Figure 1 to transform "(−)" to "(+)". For example, a higher slope is a barrier to PA access. With the transformation, the "increase" of the slope variable value (i.e., lower slope) facilitates the PA access. The variable underlined and bolded in Figure 1 indicates the new environmental variables considered at the Georgia level. Next, we ranked counties within the US and Georgia to map and integrate the 5 As index and to develop the Children's PAD index.

The variables' and 5 As indexes' ranking were normalized from 0 to 1 (min–max normalization) and then used to determine the PAD index. We used the weighted sum model from MCE [27,51] to calculate the 5 As indexes separately by the same weight of each variable within it. Next, the PAD index was a combination of 5 As indexes with the same weight of each A to generate the PAD index (Equation (1)):

$$
\begin{aligned}
PAD\ index = 1\quad &-(w_1 \times Availability\ index) + (w_2 \times Accessibility\ index)\\
&+(w_3 \times Accommodation\ index)\\
&+(w_4 \times Affordability\ index)\\
&+(w_5 \times Acceptability\ index)
\end{aligned}
\tag{1}
$$

where $w_1$ to $w_5$ are the weights applied to each one of the 5 As. We use equal weights in this study, with $w_1 = w_2 = w_3 = w_4 = w_5 = 0.2$ and $w_1 + w_2 + w_3 + w_4 + w_5 = 1$. However, it can be customized to varied weights, as long as the sum of weights is 1.

To analyze the variables in each of the 5 As, we added the environmental variables of each of the 5 As to customize Equation (1), and Equation (2) demonstrates the variables' processing within the 5 As.

$$
PAD\ index = 1 - \sum_{i=1}^{N=5} w_i \sum_{j=1}^{m} x_{ij} w_j
\tag{2}
$$

where $i$ and $w_i$ refer to the label of an A and its weight ($i$ = 1 to 5). The variables $x_{ij}$ and $w_j$ refer to an environmental variable in A$i$ following the assignment of measures and its weight in Figure 1; The variable m refers to the number of environmental variables in A$i$ (Figure 1). Higher 5 As' combination values mean better "access", and therefore,

lower values in the PAD index, which offers a more welcoming and low-disparity PA environment for children.

Last, each county's 5 As and PAD indexes were classified into three qualitative categories, Low, Average and High, using the quantile classification. To analyze the effect of the 5 As in the PAD index, we created a wind-rose visualization to show the 5 As distribution for the example counties.

### 3.5. Self-Organizing Map (SOM)

The Self-organizing map (SOM) is an unsupervised artificial neural network method for clustering and visualizing multivariable datasets. It reduces and analyzes a large dataset of a high dimension to a low dimension while retaining the data samples' internal association and pattern. We used the SOM to cluster the 5 As ranking and evaluate how those influence the children's PAD index in Georgia. We used the elbow method to determine the optimal number of clusters in the SOM [52]. The Elbow method computes the squared sum (within a cluster sum of squares) of the distance between the cluster's centroid and the sample point within the cluster based on the number of clusters [52]. Generally, a cluster with a small sum of squares is more compact than a cluster with a large sum of squares. In this study, we chose 4 clusters to analyze how 5 As vary in the PAD index (Supplementary Figure S1).

To cluster the 5 As indexes in Georgia counties, we used the *kohonen* package [53] in R. The algorithm to produce an SOM from 5 As can be summarized as follows [54,55]:

1. The 5 As were considered as five-dimensional within each county as an input vector ($5 \times 159$ matrix for 5 As and 159 counties) in an "input space".
2. Due to the elbow method, four-node weight vectors were randomly put in the space.
3. The Euclidean distance was used to calculate the best matching unit (BMU), which is the smallest distance between the node weight vectors and the input vector.
4. Update the node weight vectors and nodes' neighborhood with the BMU by pulling them closer to the input vector.

$$w_v(s+1) = w_v(s) + \theta(u,v,s) \cdot a(s) \cdot (D(t) - w_v(s)) \tag{3}$$

where $s$ is the current iteration, $v$ is the node weight vector, $u$ is the best-matching unit (BMU), $\theta(u,v,s)$ is the neighborhood function from BMU, $a(s)$ is the learning rate, and $D(t)$ is the input data vector.

5. Repeat steps 3 and 4 for N iterations.

The results were visualized in one line that showed the links between nodes and distribution by clusters. The county cluster is mapped in ArcGIS Pro to offer geographic reference.

### 4. Result

Figure 2 shows the US's 5 As indexes and children's PAD index at the county level. The 5 A "access" values from low to high shows the suitability of the physical activity environment for children from low to high from the 5 A dimensions. The PAD index value from Low to High shows the disparity of PA environment for children from Low to High. We used the US Census region and Division Map to explain the regional results (Supplementary Figure S2) [56]. The PA supply showed high availability in the divisions and subdivisions of Pacific, East North Central, Middle Atlantic, and West Florida. The low availability PA supply was shown on the edge of the West North Central, Mountain, and West South-Central Divisions. In the accessibility index, high values were in the Pacific and East North Central divisions; low accessibility was in the Eastern Center region. For the affordability index, high values were clustered in the northern US; and lower affordability values were clustered in the southern US. High accommodation values were in the South and Northeast, and low accommodation values were in the Pacific, Mountain, and East North Central divisions. High acceptability occurred in the West and Midwest divisions, and low values in the South Mountain, South and Middle Atlantic, and East South-Central

divisions. In the overall children's PAD index, the High disparity values were in Wyoming, Oregon, Wisconsin, Washington, Oregon, and New England Division; and the low PAD value were in the Deep South states and Mountain Division. Moreover, the children's PAD index could interact with other geographical classifications. Figure 3 shows the PAD index interacting with urban-to-rural classification by a bivariate choropleth map. The rural areas have the most High-disparity PAD areas; they were clustering in the East North Central, East South Central, South Atlantic, and West South-Central divisions. In urban and suburban areas, the High disparity areas were in southern California, southern Arizona, and the East North Central Division.

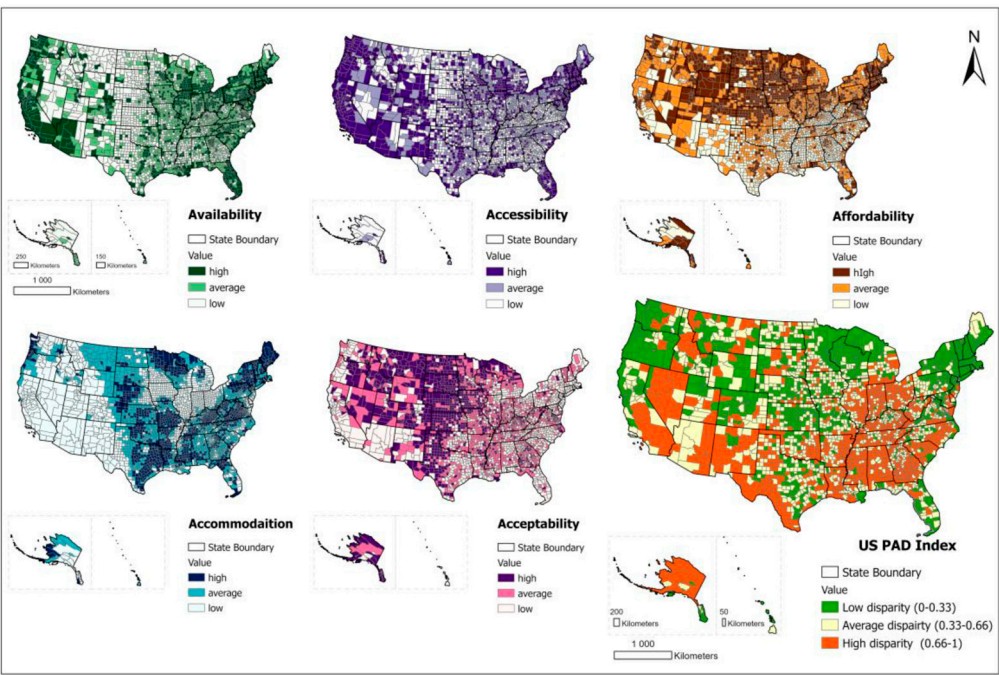

**Figure 2.** The 5 As and Children's PAD index in the US.

Figure 4 shows Georgia counties' 5 As Indexes and the children's PAD index. Compared to the US children's PAD index, Georgia's indexes included more variables (details in Figure 1) in the PAD calculation. The overall pattern of Georgia children's PAD index is similar to that of the US (Figure 2), showing high PAD values in the south. Only minor differences were observed when zooming into some geographic areas. Three of the 5 As, availability, accessibility, and affordability, showed roughly the same spatial distribution patterns. In these three As indexes, the high value counties clustered in Atlanta and its surrounding area, and the low values clustered in the South of Georgia. However, accommodation and acceptability indexes revealed an inverse pattern compared to the other three. The counties with high values were in the South of Georgia, and those with low values were in the Atlanta area. For the PAD index in Georgia, counties in the Atlanta and Coastal regions had Low disparity values indicated by the children's PAD index. However, some counties in Northeast Atlanta and most counties in South Georgia demonstrated High disparity in the children's PAD index.

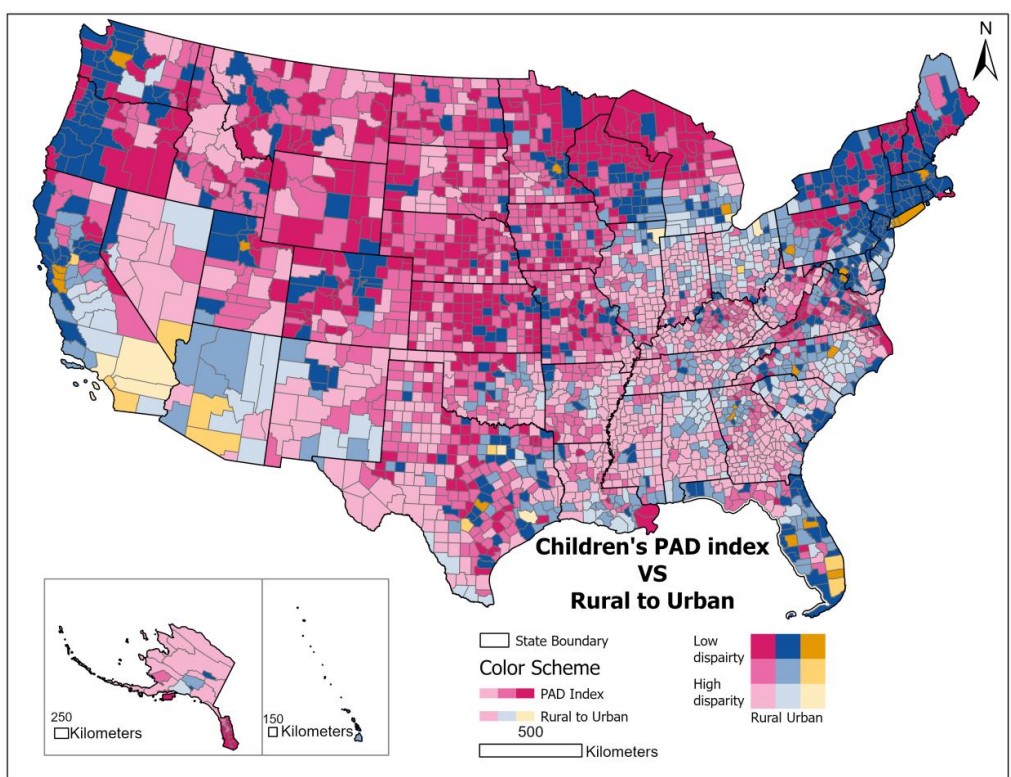

**Figure 3.** Children's PAD index vs. Rural to Urban classification in a bivariate choropleth map.

Figure 5 used wind-rose maps to show the children's PAD index measurement (5 As as five directions) with low, average, and high (from center to edge) of example counties in North Georgia–Gilmer, Hall, and Clarke; and South Georgia–Coffee, Bacon, and Chatham. There is a geographic variance of the 5 As value distribution between Georgia's South and North counties. Those counties with an Average or High value on the children's PAD index had at least one high value in one of 5 As. Gilmer County in North Georgia had a Low children's PAD index value, one low value, and no high value in the 5 As. Coffee County in South Georgia had a Low children's PAD index value, with three low values and no high values in the 5 As. This variance indicates that the 5 As impact varies by county. This variance led to further analysis to see how the 5 As distribute in each county.

To further understand how the combinations of 5 As influence the children's PAD index, we use the SOM method to analyze the Georgia counties into 4 clusters (Figure 6). This SOM result shows four combinations of 5 As in Georgia counties. Counties in the brown cluster showed a high value in availability, accessibility, and affordability, and a low value in accommodation and acceptability. Counties in the light-blue cluster showed a high value accommodation and average in the other four As. Counties in the blue cluster indicated an average value in acceptability and low in the other four As. Counties in the dark blue cluster indicated a high value in accommodation and acceptability and a low value in the other three As. Comparing the SOM and the children's PAD index in Figure 6b,c, the brown and light-blue clusters generally matched the Low and Average value counties in the PAD index. The blue and dark blue clusters generally matched High disparity in the children's PAD index. It shows that the 5 As indexes value has different contribution patterns to address the Low disparity in the PAD index. Therefore, the SOM result of 5 As values could explain what can be improved in the High disparity and what could be maintained and improve Low disparity in the children's PAD index.

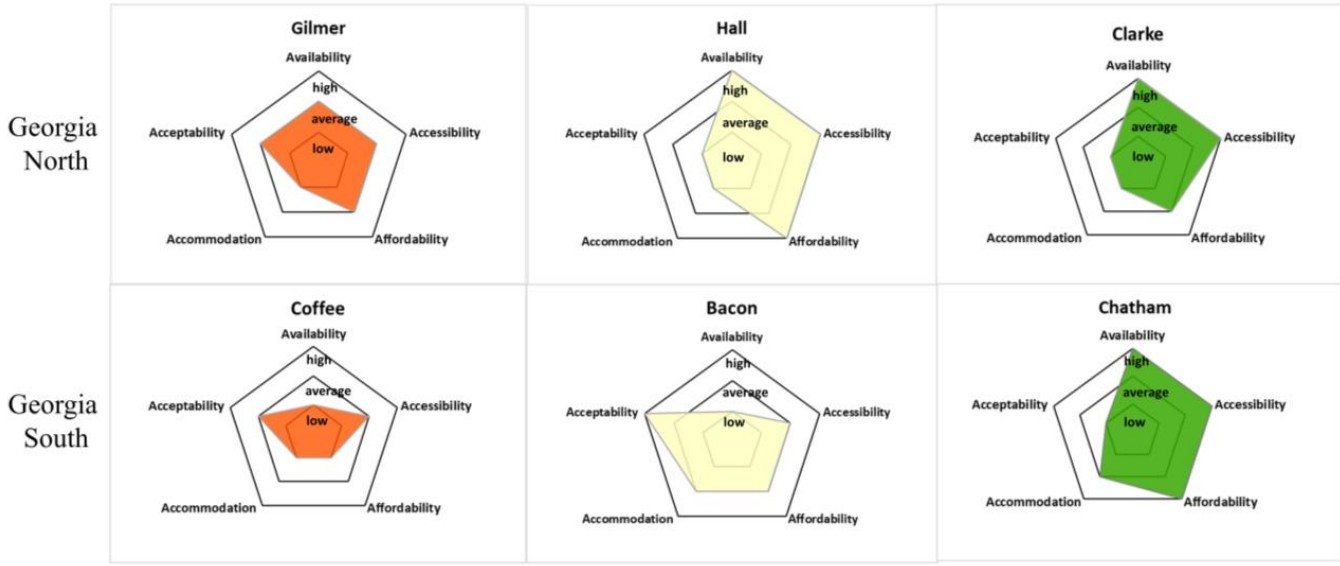

**Figure 4.** The 5 As and Children's PAD index in Georgia.

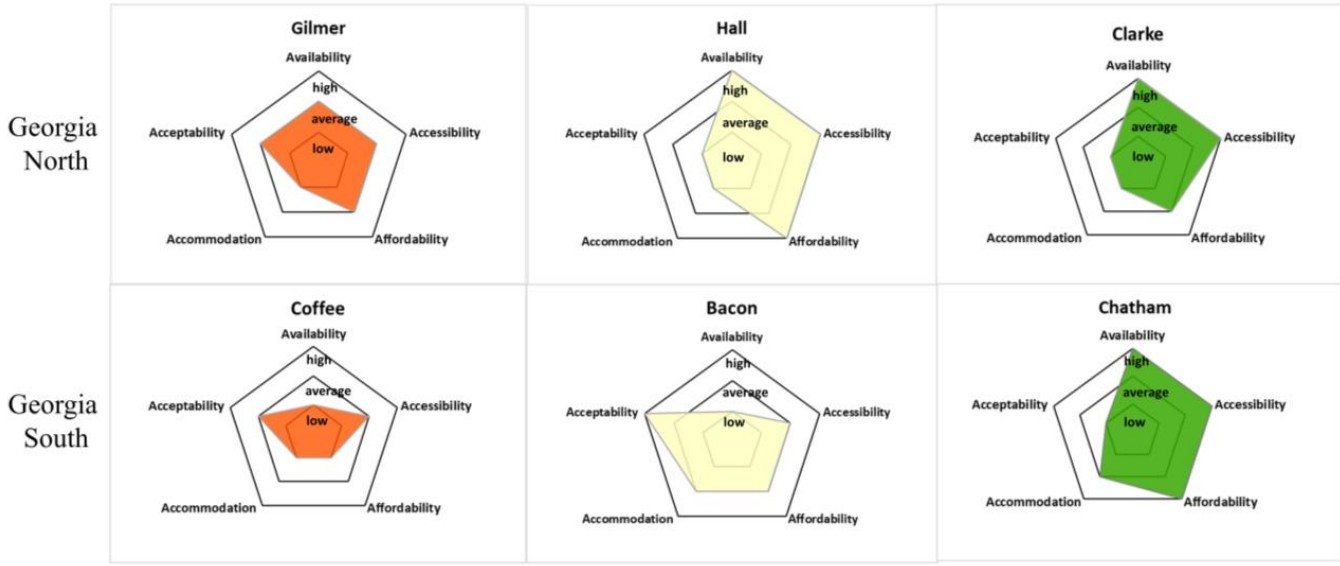

**Figure 5.** The 5 As distribution example counties in Georgia North (Gilmer, Hall, and Clarke) and South (Coffee, Bacon, and Chatham).

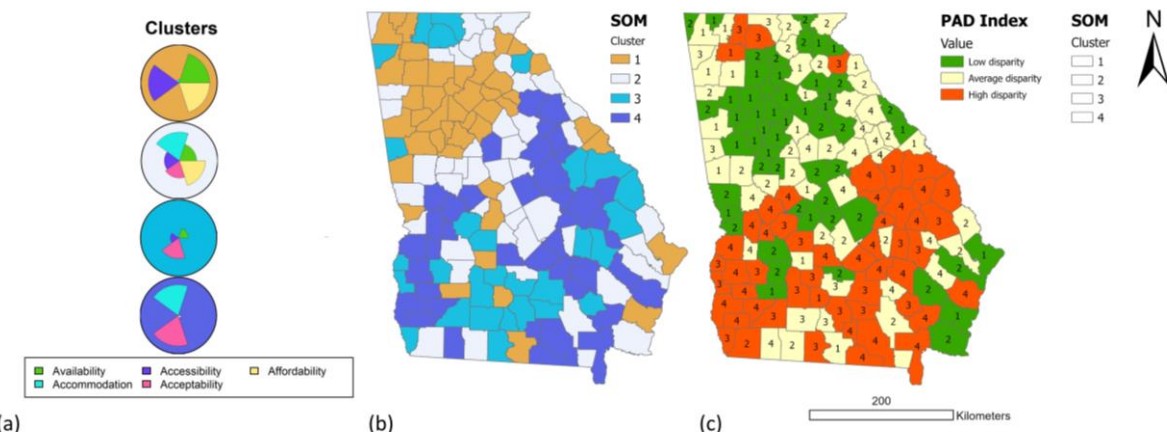

**Figure 6.** The SOM clusters compare to children's PAD index: (**a**) the cluster of 5 As ranking in Georgia counties, (**b**) SOM-cluster membership by county Georgia, (**c**) children PAD index by county in Georgia.

## 5. Discussion

### 5.1. Summary Findings

We designed and calculated the children's PAD index for the US and found regional differences. High PAD counties clustered in the US southeast and southwest (Figure 2). Different 5 As contributed to the High PAD index values in these areas. The low value of 2 As (acceptability and accommodation) and 3 As (acceptability, affordability, and accessibility) in combination contribute to High disparity values in the US southeast and southwest, respectively. The finding in the southeast and southwest region informs the policy and program directions to improve the PA environment in the south. For example, the results show a need to increase the quality of the PA environments in the US southeast and decrease the price for access to the PA environment in the US southwest. Moreover, as shown in Figure 1, there is a need for local governments and researchers to collaboratively analyze the PA disparity on a larger geographical scale (e.g., High disparity of PAD index in the Deep South states). We further explored urbanity–rurality in Figure 3 with the children's PAD index to indicate how the index interacts with other geographical classifications [57]. Most of the High PAD counties are rural, followed by suburban and urban. Additionally, the High PAD rural and urban counties were clustered in the south and west US.

The children's PAD index can also analyze the in-state PA disparity. Some states indicated a significant spatial disparity of the PAD index within the state, such as California, Texas, and Georgia. Here, we selected Georgia (Figure 5) as the case study to show the geographic variance of the children's PAD index with the 5 As indicators. The dominating 'A' influencing the children's PAD index is different between North and South Georgia, especially the low PAD index (Figures 5 and 6). The SOM result indicated that the High children PAD index counties have two combinations of the 5 As. One is an average value in the acceptability index, and the other 4 As are low values; another is a high value in the accommodation and acceptability indexes and low in the other 3 As. Different A dimensions need to be prioritized to improve the local PA disparity to improve the children's PAD. Given the interaction and association between the 5 As and the children's PAD index, we can strategically change the accommodation (water quality, air pollution), accessibility (shorten the distance to park), availability (increase the recreational places), and affordability (increase open space) in the Coastal region of Georgia, and increase accessibility, availability, and affordability in Southwest Georgia.

### 5.2. Data Collection for Children's PAD Index

The data collection process to develop the Children's PAD index is an important procedure for this study. Previous studies have explored the influence of various environmental factors on children's physical activity, built environments [58,59], natural environments [60],

and socio-demographic environments [14], either separately or in combination. However, to design and analyze the index effectively, a comprehensive framework is necessary. Existing frameworks such as the Social Vulnerability Index [61], County Health Ranking [62], or National Walkability Index [38] have integrated data related to the topics they aim to investigate. Our study involves the concept and framework from those indexes to build the children's PAD index. We collected the environmental measures from the three environment classifications, as suggested by the previous literature. Then, the different variables of the environmental measures were separated and illustrated diverse spatial outcomes in the 5 As dimensions. For example, we have considered not only the physical places that may impact children's physical activity, such as parks and recreational centers, but also the psychological factors, including people's attitudes towards these places with the Google Maps review rate. Our approach provides a comprehensive framework that enables a more nuanced and holistic understanding of the environmental factors and the data collection aspects for the future children's physical activity's study.

### 5.3. PAD Index Contributions and Future Work

This study designed and created the first Physical activity Access Disparity (PAD) index in children to analyze the PA environment suitability. Evidence from a series of studies suggested that children's PA studies should consider built, socio-demographic, and natural environmental factors [14,59,63]. We adopted GIS measurements and the 5 As of "access" conceptual framework to integrate those factors to build a multi-dimensional children's PAD index [20,26]. Combining the 5 As indexes within the children's PAD index introduces the degree of fit between the PA supply and children population demand. This index provides a new perspective on children's PA research to use the 5 As guideline to study the children's PA environment and help policymakers estimate the type of improvement for the PA environment. For example, the index could be of use for parents, school administration, park and recreation department, or the American Association of Sports Medicine, to figure out where and how to enhance, improve, and promote children's physical activity. Furthermore, with the availability of environmental variables on a local scale, this framework is flexible for a more refined analysis.

Lastly, besides the previous access studies [26], we clarify the concept of accommodation and acceptability in PA access. In our study, the accommodation dimension is objectively based on the PA environment aspect to identify how the PA environment or supply facilities accommodate the population. For example, people living with low greenness and high air pollution are associated with an increased risk of mortality, and these areas are not environmentally suitable for people to live in [35]. The acceptability dimension is according to the demand population's subjective attitude to the PA environment or facilities to define the PA supplies. It shows the safety level for children and parents to feel comfortable playing in the neighborhood due to the road traffic, crime, and vacant places [64,65].

### 5.4. Limitations

The main limitation of the children's PAD index is the data availability issue and secondary data usage. Based on the literature review, we collected data from the different databases that might impact the PA environment to build the children's PAD index. It might still not include all the possible environmental variables due to no available data at a large geographical scale level and outdated data. However, we can add more variables for future research, followed by the flexibility of the 5 As framework. For instance, we add more environmental variables to the Georgia PAD index based on the available data than the US PAD index. The second limitation is validated for the children's PAD Index. The index treated the weighted environmental variables and 5 As equally in designing the index. Future studies should weigh the variables from the advice of children, guardians, policymakers, and academics.

## 6. Conclusions

Access to PA among children has demonstrated contributions to the overall health of children. Recently, the prevalence of physical inactivity has been increasing globally, and this is concerning particularly among children due to its consequences over the course of life [1]. We propose a comprehensive public health surveillance tool to improve children's physical activity environment disparity using existing data. Our approach adopted the GIS method and the 5 As "access" concept to design and build the children's PAD index. It analyzes the geographical disparity of environmental variables in access to children's PA. The PAD index identifies "play oases" and "play deserts" for children in US and Georgia. It offers researchers and policymakers guidance to focus on the urgent area with High PAD index and analyze the 5 As situation within it to improve the PA disparity situation. Moreover, with this PAD index framework's flexible parameter-setting and weighting scheme, future work could be adapted to fit local priorities and policy/program context.

**Supplementary Materials:** The following supporting information can be downloaded at: https://www.mdpi.com/article/10.3390/ijgi12030134/s1, Figure S1: Within Cluster Sum of Squares result; Figure S2: Census Regions and Divisions of the United States (Census, 2021); Table S1: List of acronyms.

**Author Contributions:** Conceptualization, Lan Mu, Jue Yang and Janani Rajbhandari-Thapa; methodology, Lan Mu and Jue Yang; formal analysis, Jue Yang; validation, Lan Mu, Jue Yang and Janani Rajbhandari-Thapa; software, Jue Yang; writing—original draft preparation, Jue Yang and Lan Mu; writing—review and editing, Jue Yang, Lan Mu and Janani Rajbhandari-Thapa; visualization, Jue Yang Funding acquisition, Lan Mu and Janani Rajbhandari-Thapa. All authors have read and agreed to the published version of the manuscript.

**Funding:** Partially funded by UGA President's Interdisciplinary Seed Grant Program Impact of the School and Surrounding Environment on Implementation of Georgia's Statewide Childhood Obesity Policy.

**Data Availability Statement:** All data available is in the Supplementary Materials reference.

**Conflicts of Interest:** The authors declare no conflict of interest.

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
