# Peer review of "Measuring and Mapping Physical Activity Disparity (PAD) Index Based on Physical Activity Environment for Children"

_ijgi, doi:10.3390/ijgi12030134_

Round 1

Reviewer 1 Report

The Abstract section presents in a quite clear manner the scope as well as the methodological framework of the paper. The Introduction section presents the important concepts of the theoretical framework as well as it summarizes the niche in which this paper responds. My suggestion would be to include in this section, a short description of paper’s sections as well as the research questions. The first paragraph on the 2.2 subsection needs more justification based on literature findings. My suggestion would be also authors to add in the Data and Method section the exact research questions as well as the research tools used. Variables could be presented in a table as well as their important characteristics and then in a paragraph  authors could analyse them. The Results and Discussion section could be longer so as to further present and analyse important data collected, compared also with modern literature.

Author Response

Please find the response document in the attachment.

Thank you for the review!

Reviewer 2 Report

I find the research presented in the manuscript relevant and practical in its approach. The authors use a complex GIS methodology in their study. And I appreciate the fact that they conduct their GIS analysis not only for the Georgia area but also for the whole USA.

However, the manuscript has some shortcomings in both content and form, I assume. I suppose that the presentation of the research is a bit ‘US-centric’. At the same time, given the global nature of the IJGI, it should go into more detail, in my opinion, on some details that may be less clear to a non-US reader. I detail what I mean below:

a)       "We used 13 miles, the average travel distance to recreation destinations for both the US and Georgia, according to the National Household Travel Survey [34], to define the area for the PA supply count" (p. 4.). Could you briefly justify in the body text, and not just by citing the source, please, why this is 13 miles? And a formal point of mine is that whenever you mention miles, the metric equivalent of the imperial system should also be indicated, for example in parentheses. Also, in the case of the metric version of the scale on the maps, I suppose that the metric version should also be indicated.

b)      The manuscript also contains the shortcoming, which occurs in GIS publications, that authors already perform complex statistical calculations, but at the same time they do not adequately state how accurate their input data sources are and what their data mean in the given geographical context (in this case, the USA and Georgia).

Thus, concerning the results of their more complex statistical calculations, it may remain somewhat unclear what exactly they refer to, and what authors can or cannot measure by them. Based on that, I would be interested to know if you could explain the input variables below just briefly for the sake of non-US readers.

c)       National walkability Index

d)      National Risk Index

e)      The rate of population below the poverty level (here it would be useful to describe the national average value)

f)        No High School Diploma (by High School Diploma more precisely what you mean)

g)       Nearest Distance to the School (what school, what age group, according to reservations, public or private)?

h)      Nearest Distance to the Park (since in the USA there may be parks with semi- or full private legal status, which can also affect accessibility, I would be interested to know if you have any relevant statistics on this aspect of the parks)

i)        Income

At the end of the manuscript, the authors could make slightly more and more specific policy recommendations based on their research.

Some figures/maps in the manuscript (Fig. 1, p. 5., Fig. 2., p. 7, Fig. 5., p. 10.) are quite difficult to interpret, in my opinion, as they are relatively small and of poor resolution, and therefore they should be enhanced.

Author Response

Please find the response in the attachment.

Thank you for the review!

Reviewer 3 Report

I have reviewed the manuscript. First of all I appreciate the authors thinking to make this work successful.

The manuscript is written with care.

The methodology is well designed.

This is a novel work where the authors collect data from target groups and analyse the findings.

The major limitations were also highlighted by authors.

Conclusion part is very informative.

So, I recommend for accepting this work to publish in your reputed journal only after checking the grammatical and sentence errors in some places.

Thank you.

Author Response

(The authors gave the same response as above.)
